# Learning Object-Language Alignments for Open-Vocabulary Object Detection

**Chuang Lin**[1]* **Peize Sun**[3] **Yi Jiang**[2] **Ping Luo**[3] **Lizhen Qu**[1]
**Gholamreza Haffari**[1] **Zehuan Yuan**[2] **Jianfei Cai**[1]
[1] Monash University [2] ByteDance [3] The University of Hong Kong

## Abstract

Existing object detection methods are bounded in a fixed-set vocabulary by costly labeled data. When dealing with novel categories, the model has to be retrained with more bounding box annotations. Natural language supervision is an attractive alternative for its annotation-free attributes and broader object concepts. However, learning open-vocabulary object detection from language is challenging since image-text pairs do not contain fine-grained object-language alignments. Previous solutions rely on either expensive grounding annotations or distilling classification-oriented vision models. In this paper, we propose a novel open-vocabulary object detection framework directly learning from image-text pair data. We formulate object-language alignment as *a set matching problem* between a set of image region features and a set of word embeddings. It enables us to train an open-vocabulary object detector on image-text pairs in a much simple and effective way. Extensive experiments on two benchmark datasets, COCO and LVIS, demonstrate our superior performance over the competing approaches on novel categories, e.g. achieving 32.0% mAP on COCO and 21.7% mask mAP on LVIS. Code is available at: `https://github.com/clin1223/VLDet`.

## 1 Introduction

In the past few years, significant breakthroughs of visual models have been witnessed on close-set recognition, where the object categories are pre-defined by the dataset (Johnson-Roberson et al., 2016; Sakaridis et al., 2018; Geiger et al., 2013; Yu et al., 2018). Such achievements are not the final answer to artificial intelligence, since human intelligence can perceive the world in an open environment. This motivates the community moving towards the open-world setting, where visual recognition models are expected to recognize arbitrary novel categories in a zero-shot manner. Towards this goal, learning visual models from language supervisions becomes more and more popular due to their open attributes, rich semantics, various data sources and nearly-free annotation cost. Particularly, many recent image classification works (Yu et al., 2022; Yuan et al., 2021; Zhai et al., 2021; Jia et al., 2021; Radford et al., 2021; Zhou et al., 2021a; Rao et al., 2021; Huynh et al., 2021) successfully expand their vocabulary sizes by learning from a large set of image-text pairs and demonstrate very impressive zero-shot ability. Following the success in open-vocabulary image classification, a natural extension is to tackle the object detection task, i.e., open-vocabulary object detection (OVOD).

Object detection is a fundamental problem in computer vision aiming to localize object instances in an image and classify their categories (Girshick, 2015; Ren et al., 2015; Lin et al., 2017; He et al., 2017; Cai & Vasconcelos, 2018). Training an ordinary object detector relies on manually-annotated bounding boxes and categories for objects (Fig. 1 Left). Similarly, learning object detection from language supervision requires object-language annotations. Prior studies investigated using such annotations for visual grounding tasks (Krishna et al., 2017; Yu et al., 2016; Plummer et al., 2015; Zhuang et al., 2018). Most of the existing open-vocabulary object detection works (Li et al., 2021; Kamath et al., 2021; Cai et al., 2022) depend entirely or partially on the grounding annotations, which however is not scalable because such annotations are even more costly than annotating object detection data. To reduce the annotation cost for open-vocabulary object detection, a handful of

---

*This work was performed while Chuang Lin (chuang.lin@monash.edu) worked as an intern at ByteDance.

Figure 1: Left: conventional object detection. Right: our proposed open-vocabulary object detection, where we focus on using the corpus of image-text pairs to learn region-word alignments for object detection. By converting the image into a set of regions and the caption into a set of words, the region-word alignments can be solved as a *set-matching problem*. In this way, our method is able to directly train the object detector with image-text pairs covering a large variety of object categories.

recent studies (Du et al., 2022; Gu et al., 2021) distill visual region features from classification-oriented models by cropping images. However, their performance is limited by the pre-trained models, trained based on global image-text matching instead of region-word matching.

In this paper, we propose a simple yet effective end-to-end vision-and-language framework for open-vocabulary object detection, termed **VLDet**, which directly trains an object detector from image-text pairs without relying on expensive grounding annotations or distilling classification-oriented vision models. Our key insight is that extracting region-word pairs from image-text pairs can be formulated as *a set matching problem* (Carion et al., 2020; Sun et al., 2021b;a; Fang et al., 2021), which can be effectively solved by finding a bipartite matching (Kuhn, 1955) between regions and words with minimal global matching cost. Specifically, we treat image region features as a set and word embedding as another set with the the dot-product similarity as region-word alignment score. To find the lowest cost, the optimal bipartite matching will force each image region to be aligned with its corresponding word under the global supervision of the image-text pair. By replacing the classification loss in object detection with the optimal region-word alignment loss, our approach can help match each image region to the corresponding word and accomplish the object detection task.

We conduct extensive experiments on the open-vocabulary setting, in which an object detection dataset with localized object annotations is reserved for base categories, while the dataset of image-text pairs is used for novel categories. The results show that our method significantly improves the performance of detecting novel categories. On the open-vocabulary dataset COCO, our method outperforms the SOTA model PB-OVD (Zhou et al., 2022) by 1.2% mAP in terms of detecting novel classes using COCO Caption data (Chen et al., 2015). On the open-vocabulary dataset LVIS, our method surpasses the SOTA method DetPro (Du et al., 2022) by 1.9% $mAP^{mask}$ in terms of detecting novel classes using CC3M (Sharma et al., 2018) dataset.

The contributions of this paper are summarized as follows: 1) We introduce an open-vocabulary object detector method to learn object-language alignments directly from image-text pair data. 2) We propose to formulate region-word alignments as a set-matching problem and solve it efficiently with the Hungarian algorithm (Kuhn, 1955). 3) Extensive experiments on two benchmark datasets, COCO and LVIS, demonstrate our superior performance, particularly on detecting novel categories.

## 2 RELATED WORK

**Weakly-Supervised Object Detection (WSOD).** WSOD aims to train an object detector using image-level labels as supervision without any bounding box annotation. Multi-instance learning (Dietterich et al., 1997) is a well studied strategy for solving this problem (Bilen et al., 2015; Bilen & Vedaldi, 2016; Cinbis et al., 2016). It learns each category label assignment based on the top-scoring strategy, i.e. assigning the top-scoring proposal to the corresponding image label. As no bounding box supervision in training, OICR (Tang et al., 2017) proposes to refine the instance classifier online using spatial relation, thereby improving the localization quality. Cap2Det (Ye et al., 2019) further utilizes a text classifier to convert captions into image labels. Based on the student-teacher framework, Omni-DETR (Wang et al., 2022) uses different types of weak labels to generate accurate pseudo labels through a bipartite matching as filtering mechanism. Different from WSOD, OVOD

benefits from the fully-annotated base class data, resulting in accurate proposals to match with corresponding words for target classes not known in advance. Directly using WSOD methods for OVOD tends to assign a region proposal to the mostly matching class, which is likely to be a base class due to the full supervision for the bass classes.

**Multi-Modal Object Detection.** A recent line of work considers instance-wise vision-language tasks, where the free-form language from captions is required to align with the objects. GLIP (Li et al., 2021) unifies object detection and visual grounding into a grounded language-image pre-training model. MDETR (Kamath et al., 2021) develops an end-to-end text-modulated detection system derived from DETR (Carion et al., 2020). XDETR (Cai et al., 2022) independently trains the two streams, visual detector and language encoder, and align the region features and word embeddings via dot product operations. All these multi-modal detectors extend their vocabulary fully or partially relying on annotated grounding data, i.e. object-language pairs with ground truth bounding boxes. In contrast, our model directly learns the region-word alignments for the novel classes by bipartite matching from the image-text pairs without bounding box annotations.

**Open-Vocabulary Object Detection (OVOD).** Recently, there is a trend toward improving the generalization ability of object detectors to new categories via multi-modal knowledge. Open-vocabulary object detection aims to increase the vocabulary of object detection with image-text pairs. As the first work of OVOD, OVR-CNN (Zareian et al., 2021) uses a corpus of image-caption pairs to pretrain a vision-language model and transfers the visual backbone to the supervised object detector. With the huge progress in image-level open vocabulary recognition (Yu et al., 2022; Yuan et al., 2021; Zhai et al., 2021; Jia et al., 2021; Radford et al., 2021), distilling the knowledge from the off-the-shelf vision-language model like CLIP (Radford et al., 2021) has become a popular solution for OVOD tasks (Zhong et al., 2022; Minderer et al., 2022). For instance, Zhong et al. (2022) and Gao et al. (2021) generate the pseudo region annotations from the pre-trained vision-language model and use them as training data for detectors. To get a better localization ability for generating pseudo region annotations, Zhao et al. (2022) use another two-stage detector as a strong region proposal network (RPN). In order to use the CLIP class embedding effectively, DePro (Du et al., 2022) designs a fine-grained automatic prompt learning. Different from these works, we scale up the vocabulary of object detection by using the annotation-free data of image-text pairs. *As it is easy to extract noun words from captions, our object vocabulary is easily to be extended and scaled.* Besides, some work focuses on improving image segmentation in open vocabulary settings. Ghiasi (Ghiasi et al., 2022) organizes pixels into groups first and then learns the alignments between captions and predicted masks. Another close work is Detic (Zhou et al., 2022) that focuses on increasing the performance on novel classes with image classification data by supervising the max-size proposal with all image labels. Unlike Detic, we try to align image regions and caption words by bipartite matching.

## 3 METHOD

### 3.1 OVERVIEW

A conventional object detector can be considered as trained with $(o_i, g_i)$, where $o_i$ denotes the $i\text{-}th$ object/region/box feature of an input image $I$, and $g_i = (b_i, c_i)$ denotes the corresponding labels with the bounding-box coordinates $b_i$ and their associated categories $c_i \in \mathbb{C}^{base}$, where $\mathbb{C}^{base}$ is the set of base classes, or pre-defined classes. Open-vocabulary Object Detection (OVOD) (Zareian et al., 2021) aims to acquire a large vocabulary of knowledge from a corpus of image-caption pairs to extend detection of pre-defined classes to novel classes. In other words, our goal is to build an object detector, trained on a dataset with base-class bounding box annotations and a dataset of image-caption pairs $\langle I, C \rangle$ associated with a large vocabulary $\mathbb{C}^{open}$, to detect objects of novel classes $\mathbb{C}^{novel}$ during testing. Note that the vocabulary $\mathbb{C}^{base}$ and $\mathbb{C}^{novel}$ might or might not overlap with $\mathbb{C}^{open}$.

Fig. 2 gives an overview of our proposed VLDet framework. We propose to learn fine-grained region-word alignments with the corpus of image-text pairs, which is formulated as a set-matching problem, i.e., matching a set of object candidates with a set of word candidates. In the following, we first describe how to formulate the learning of region-word alignment from image-text pairs as a bipartite match problem in Section 3.2 and then depict the network architecture followed by some method details in Section 3.3.

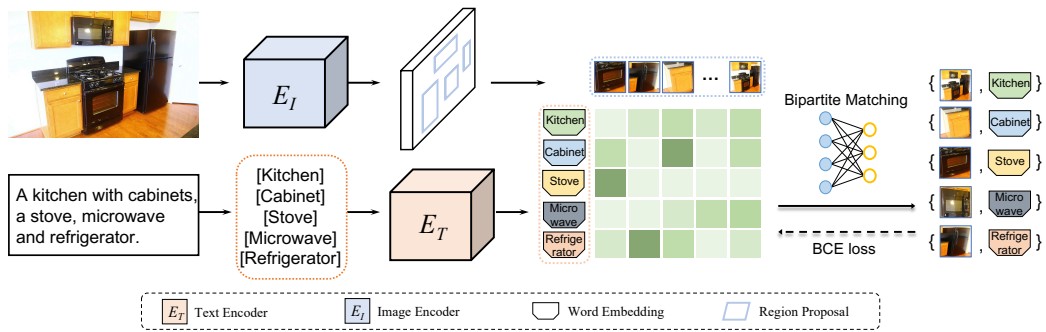

Figure 2: Overview of our proposed **VLDet** framework. We propose to learn fine-grained region-word alignments with the corpus of image-text pairs, which is formulated as a bipartite matching problem between image regions and word candidates.

## 3.2 LEARNING OBJECT-LANGUAGE ALIGNMENTS BY BIPARTITE MATCHING

**Recap on Bipartite Matching.** The Bipartite Matching describes the following problem: supposing there are $X$ workers and $Y$ jobs. Each worker has a subset of jobs that he/she is capable to finish. Each job can only accept one worker and each worker can be appointed to only one job. Because each worker has different skills, the cost $d_{x,y}$ required to perform a job $y$ depends on the worker $x$ who is assigned to it. The goal is to determine the optimum assignment $M^*$ such that the total cost is minimized or the team effectiveness is maximized. For this purpose, given a matching matrix $\mathbf{M}$, if element $m_{i,j} = 1$, it indicates a matching; otherwise, not matching. We can formulate this matching problem as:

$$\min_{\mathbf{M}} \sum_{i=1}^{X} \sum_{j=1}^{Y} d_{i,j} m_{i,j} \tag{1}$$

The constraints is to ensure each job being assigned to one worker if there are more workers; otherwise, ensure each worker being assigned to one job. In our case, the jobs are object regions $\mathbf{r}_i$ from image $I$ and the workers are words $\mathbf{w}_j$ from caption $C$.

**Learning Object-Language Alignments from Image-Text Pairs.** Because the image-text pairs data is easy to acquire, we exploit it to sharply increase the coverage of object classes by leveraging the large vocabulary from an image-text dataset. However, critical information is missing for the existing object detection models, which require labeling of image regions with words for training. Hence, the key challenge is to find the correspondences between the set of regions and the set of words. Instead of generating pseudo bounding boxes and labels for each image, we propose to formulate the problem of region-word alignments as an optimal bipartite matching problem.

Given an image $I$, the candidate region features from the image encoder is defined as $\mathbf{R} = [\mathbf{r}_1, \mathbf{r}_2, \ldots, \mathbf{r}_m]$, where $m$ is the number of the regions. Given a caption $C$, we extract all the nouns from the caption and embed each noun with the language encoder. The word embedding is defined as $\mathbf{W} = [\mathbf{w}_1, \mathbf{w}_2, \ldots, \mathbf{w}_{|\mathbf{W}|}]$, where the $|\mathbf{W}|$ is the number of nouns in the caption. Usually, we have $m > |\mathbf{W}|$, as the region proposal network provides sufficient candidate regions. We define each region as a "job" which tries to find the most suitable "worker" and each word as a "worker" who finds the most confident "job". In this context, our approach converts the region and word assignment task into a set-to-set bipartite matching problem from a global perspective. The cost between image regions and words is defined as the alignment scores $\mathbf{S} = \mathbf{W}\mathbf{R}^\top$. The bipartite matching problem can then be efficiently solved by the off-the-shelf Hungarian Algorithm (Kuhn, 1955). After matching, the classifier head of the detection model is optimized by the following cross-entropy loss:

$$L_{region-word} = \sum_{i=1}^{|W|} - \left[ \log \sigma(s_{ik}) + \sum_{j \in W'} log(1 - \sigma(s_{jk})) \right], \tag{2}$$

where $\sigma$ is the sigmoid activation, $s_{ik}$ is the alignment score of the $i$-th word embedding and its corresponding $k$-th region feature matched by the bipartite matching, and $W'$ is the set of nouns from other captions in the same batch.

Besides, we further consider image-text pairs as special region-word pairs. Particularly, we extract the RoI feature of an image by considering the entire image as a special region and the entire caption feature from the text encoder as a special word. For an image, we consider its caption as a positive sample and other captions in the same minibatch as negative samples. We use a similar binary cross-entropy loss $L_{image-text}$ for image-text pairs as Eq. 2.

**Object Vocabulary.** The object vocabulary during the training can be the object labels $\{\mathbb{C}^{base}, \mathbb{C}^{novel}\}$ defined in the dataset, as many recent works (Zhou et al., 2022; Gao et al., 2021) do for LVIS and COCO. However, we notice that it is not strictly following the open-vocabulary setting. In our design, we set the object vocabulary as all nouns in the image captions in each training batch. From the perspective of the whole training process, our object vocabulary $\mathbb{C}^{open}$ size is much larger than the dataset label space. Our experiments show that this setting not only realizes the desirable open-vocabulary detection, but also achieves better performance than previous works. More details are discussed in the experiment section.

### 3.3 NETWORK ARCHITECTURE

Our VLDet network includes three components: a visual object detector, a text encoder, and an alignment between regions and words. We choose the two-stage framework Faster R-CNN (Ren et al., 2015; Cai & Vasconcelos, 2018; Zhou et al., 2021b) as our object detector component. The first stage in the object detector remains the same as Faster R-CNN, predicting object proposals by the region proposal network. To adapt the two-stage object detector into the open-vocabulary setting, we modify the second-stage in two aspects: (1) we use the class-agnostic localization head instead of the class-specific one. In this way, The localization head predicts bounding boxes regardless of their categories. (2) We replace the trainable classifier weights with the language embeddings to convert the detector to the open-vocabulary setting. We use a pretrained language model CLIP (Radford et al., 2021) as our text encoder. Noting that there is no specific design in the object detector architecture, it is easy to replace Faster R-CNN by any other detector like Transformer-based detectors (Carion et al., 2020; Zhu et al., 2020; Meng et al., 2021; Zhang et al., 2022).

**Training and Inference.** The training data consists of a dataset annotated with base-classes a dataset of image-text pairs. On the dataset with base-classes, the model is trained in a classical two-stage detection pipeline including classification and localization. For each image-caption pair, the image encoder generates candidate regions by taking the image as input and the text encoder encodes nouns in the caption as word embeddings. After that, the region features and word embeddings are aligned by the bipartite matching, followed by optimizing the model parameters. During inference, VLDet adopts the inference process of the two-stage object detector by incorporating the language embeddings into the classifier.

## 4 EXPERIMENTS

### 4.1 DATASETS

**COCO and COCO Caption.** Following open-vocabulary COCO setting (OV-COCO) (Zareian et al., 2021), the COCO-2017 dataset is manually divided into 48 base classes and 17 novel classes, which are proposed by the zero-shot object detection (Bansal et al., 2018). We keep 107,761 images with base class annotations as the training set and 4,836 images with base and novel class annotations as the validation set. For images-text pairs data, we use COCO Caption (Chen et al., 2015) training set, which contains 5 human-generated captions for each image. Following (Gu et al., 2021; Zareian et al., 2021), we report mean Average Precision (mAP) at an IoU of 0.5.

**LVIS and Conceptual Captions.** To study a large-scale generalized setting for open-vocabulary object detection, we conduct experiments on the LVIS (Gupta et al., 2019) benchmark with Conceptual Captions (CC3M) (Sharma et al., 2018) as the paired image-text training data. LVIS dataset with object detection and instance segmentation annotations has diverse categories, which is more suitable for the open-vocabulary object detection task. In this setting, we follow ViLD (Gu et al., 2021) with the common classes and frequency classes as base classes (866 categories) and rare classes as novel classes (337 categories). We remove novel class annotations in training and predict all categories in testing. Different from COCO Caption (Chen et al., 2015) using the same images as COCO-2017, CC3M (Sharma et al., 2018) collects 3 million image-text pairs from the web. Following the official LVIS setting, we train a mask head on base-class detection data and report the mask AP for all categories.

Table 1: Open-vocabulary object detection results on COCO dataset. We follow the OVR-CNN (Zareian et al., 2021) using the same training data of COCO Caption (Chen et al., 2015) and outperform the state-of-art method PB-OVD (Gao et al., 2021) on novel classes.

| Method | Novel AP | Base AP | Overall AP |
|---|---|---|---|
| Base-only | 1.3 | 52.8 | 39.3 |
| OVR-CNN (Zareian et al., 2021) | 22.8 | 46.0 | 39.9 |
| Detic (Zhou et al., 2022) | 27.8 | 47.1 | 42.0 |
| RegionCLIP (Zhong et al., 2022) | 26.8 | 54.8 | 47.5 |
| ViLD (Gu et al., 2021) | 27.6 | 59.5 | 51.3 |
| PB-OVD (Gao et al., 2021) | 30.8 | 46.1 | 42.1 |
| Our | **32.0** | 50.6 | 45.8 |

Table 2: Open-vocabulary object detection results on LVIS dataset for two different backbones ResNet50 (RN50) (He et al., 2016) and Swin-B (Liu et al., 2021). Base-only method only uses base class bounding box, which is considered as our baseline.

| Method | Backbone | $mAP^{mask}_{Novel}$ | $mAP^{mask}_c$ | $mAP^{mask}_f$ | $mAP^{mask}_{all}$ |
|---|---|---|---|---|---|
| Base-only | RN50 | 16.3 | 31.0 | 35.4 | 30.0 |
| ViLD (Gu et al., 2021) | RN50 | 16.6 | 24.6 | 30.3 | 25.5 |
| RegionCLIP (Zhong et al., 2022) | RN50 | 17.1 | 27.4 | 34.0 | 28.2 |
| DetPro (Du et al., 2022) | RN50 | 19.8 | 25.6 | 28.9 | 25.9 |
| Detic (Zhou et al., 2022) | RN50 | 19.5 | - | - | 30.9 |
| Our | RN50 | **21.7** | 29.8 | 34.3 | 30.1 |
| Base-only | Swin-B | 21.9 | 40.5 | 43.3 | 38.4 |
| Detic (Zhou et al., 2022) | Swin-B | 23.9 | 40.2 | 42.8 | 38.4 |
| Our | Swin-B | **26.3** | 39.4 | 41.9 | 38.1 |

## 4.2 IMPLEMENTATION DETAILS

In each mini-batch, the ratio of base-class detection data and image-text pair data is 1:4. For better generalized ability, we use fixed CLIP text encoder to embed the caption and object words. To reduce training time, we always initialize the parameters from the detector trained by the fully supervised base-class detection data following (Zhou et al., 2022).

In the open-vocabulary COCO experiments, we basically follow the OVR-CNN (Zareian et al., 2021) setting without any data augmentation. Our model adopts Faster R-CNN (Ren et al., 2015) with ResNet50-C4 (He et al., 2016) as the backbone. For the warmup, we increase the learning rate from 0 to 0.002 for the first 1000 iterations. The model is trained for 90,000 iterations using SGD optimizer with batch size 8 and the learning rate is scaled down by a factor of 10 at 60,000 and 80,000 iterations.

In open-vocabulary LVIS experiments, we follow Detic (Zhou et al., 2022) to adopt Center-Net2 (Zhou et al., 2021b) with ResNet50 (He et al., 2016) as backbone. We use large scale jittering (Ghiasi et al., 2021) and repeat factor sampling as data augmentation. For the warmup, we increase the learning rate from 0 to 2e-4 for the first 1000 iterations. The model is trained for 90,000 iterations using Adam optimizer with batch size 16. All the expriments are conducted on 8 NVIDIA V100 GPUs.

## 4.3 OPEN-VOCABULARY COCO

Table 1 shows the performance comparisons of different methods for open-vocabulary COCO datasets. It can be seen that our model performs the best on novel classes, suggesting the superiority of using the bipartite matching loss with the image-text pairs. Base-only method indicates the Faster R-CNN trained with the fully supervised COCO base-category detection data with CLIP embeddings as the classifier head. Although CLIP has the generalized ability to the novel class, it only achieves 1.3 mAP. Although ViLD and RegionCLIP distills the region proposal features using the CLIP, their performances are inferior to ours on novel categories, which is the main metric in the open-vocabulary object detection setting. These distillation methods require both the image encoder and the text encoder from the pretrained CLIP model to learn the matching between the

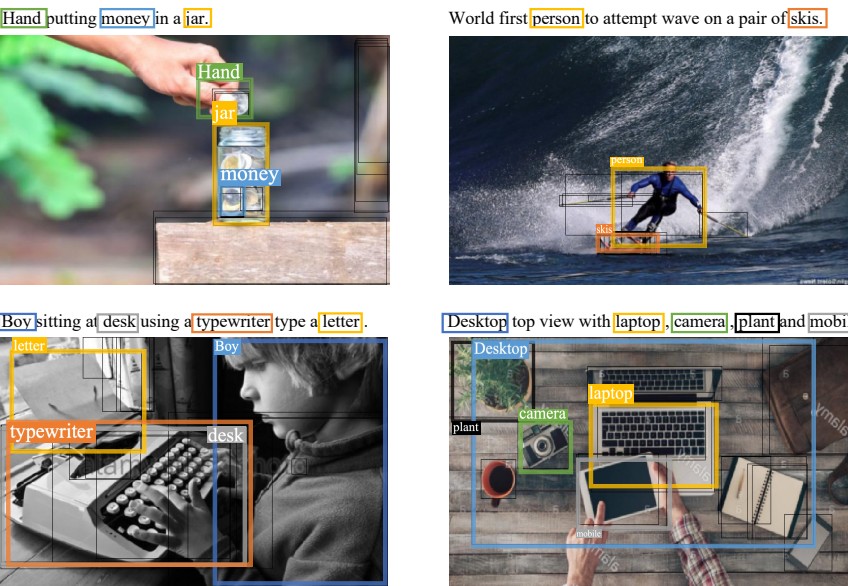

Figure 3: Visualization of the bipartite matching results. Here we show how the proposed region matched with their corresponding words. Best viewed on screen.

image regions and the vocabulary. Thus, their performances on novel classes are limited by the pre-trained model, which is trained for global image-text matching instead of region-word matching. Compared with the state-of-the-art method PB-OVD (Gao et al., 2021) which leverages the COCO Caption (Chen et al., 2015), VG (Krishna et al., 2017) and SBU Caption dataset (Ordonez et al., 2011) with about 1M training image-text pairs, our method still outperforms it on all metrics while we use only 10K images of COCO Caption as training data.

## 4.4 OPEN-VOCABULARY LVIS

Table 2 shows the performance comparisons on the open-vocabulary LVIS object detection. The "Based-only" method is only fully supervised on the base categories, which is the baseline of our method. As mentioned above, since we use fix CLIP embeddings as the classifier head, the base-only method has a certain generalization ability. The proposed model outperforms the state-of-the-art method DetPro (Du et al., 2022) by 1.9 % AP on novel classes. RegionCLIP (Zhong et al., 2022) uses a CLIP model to predict pseudo-labels for image regions and pretrains a visual model with the region-text pairs. We can see that it is effective to improve the generalization ability to novel classes. Similar strategy is used in recent works (Gao et al., 2021). However, such two-step pseudo-label method is easy to amplify false predictions and it relies on good initialization of the teacher model to generate good pseudo-labels. In contrast, our proposed method is assigning class labels from a global perspective and optimizing the object detection model without finetuning. To further explore the scalability of our model, we use Swin-B as our backbone, and our model outperforms the ResNet backbone with 4.5% mAP on novel classes. Compared with Detic, our model achieves 2.4% gain on OV-LVIS novel classes, suggesting the effectiveness of the label assignment from the global perspective.

**Visualization.** As shown in Figure 3, we visualize some cases of matching results of CC3M in open vocabulary LVIS setting. As we can see, the model can extract promising region-words pairs from image-text data, avoiding expensive annotations. The results of the matching include a variety of objects, such as "jar", "typewriter", and "camera", which indicates that our method significantly expands the vocabulary for object detection. It demonstrates the generalization ability of our method to novel classes.

## 4.5 ABLATION STUDIES

**Object Vocabulary Size.** In our VLDet, we follow the previous work (Zhong et al., 2022) using all nouns in COCO Caption and CC3M and filtering out low-frequency words, resulting in 4764/6250

Table 3: Ablation study on vocabulary size. We replace our object vocabulary with the category names of OV-COCO and OV-LVIS. The results show that training with a large vocabulary size enables better generalization.

| Vocabulary | OV-COCO | | | OV-LVIS | | |
|---|---|---|---|---|---|---|
| | Size | $mAP_{novel}$ | $mAP_{all}$ | Size | $mAP^{mask}_{novel}$ | $mAP^{mask}_{all}$ |
| Category names | 65 | 28.2 | 42.8 | 1203 | 18.9 | 31.2 |
| All nouns from caption | 4764 | 30.0 | 44.6 | 6750 | 20.4 | 30.4 |

Table 4: Ablation study on matching strategy. We implement Hungarian algorithm (Kuhn, 1955) for one-to-one region-word matching and Sinkhorn algorithm (Cuturi, 2013) for one-to-many matching.

| Matching Strategy | OV-COCO | | OV-LVIS | |
|---|---|---|---|---|
| | $mAP_{novel}$ | $mAP_{all}$ | $mAP^{mask}_{novel}$ | $mAP^{mask}_{all}$ |
| One-to-Many | 29.1 | 44.6 | 18.5 | 29.2 |
| One-to-One | 32.0 | 45.8 | 21.7 | 30.1 |

concepts left. We notice that many works recently use the LVIS and COCO label spaces as the object vocabulary during training, which do not strictly follow the open-vocabulary setting. We further analyze the effect of training our model with different vocabulary sizes. In Table 3, we replace our object vocabulary with the category names in COCO and LVIS datasets, i.e. only using the category names in the captions. We can see from Table 3 that a larger vocabulary significantly boosts the unseen class performance. It achieves gains of 1.8% and 1.5% on the novel categories of OV-COCO and OV-LVIS, respectively, indicating that training with a large vocabulary size leads to better generalization. In other words, with a bigger vocabulary, the model can learn more object language alignments which benefits the novel-class performance during inference.

**One-to-One vs. One-to-Many.** The key to extract a set of image region-word pairs from an image-text pair is to optimize the assignment problem from a global perspective. To further investigate the impact of assignment algorithms, we implement two global algorithms, Hungarian (Kuhn, 1955) and Sinkhorn (Cuturi, 2013) algorithms, where the former does one-to-one region-word assignment, and the latter provides soft alignments between one word and many regions. Considering that there may be multiple instances of the same category in an image, Sinkhorn algorithm is able to produce multiple regions to the same word, while it might also introduce more noisy region-word pairs. From Table 4, we observe that the one-to-one assignment achieves 32.0 AP and 21.7 AP for novel classes, both outperforming the one-to-many assignments of 29.1 AP and 18.5 AP. The one-to-one assignment assumption sharply reduces mis-alignments by providing each word with a high-quality image region.
*Discussion on noisy image-text pairs.* In VLDet, we employ the Hungarian algorithm to assign each word a unique region. When dealing with multi-word expressions in a text, such as "a basket of oranges", we select the most confident region-word pair and ignore the remaining ones, such as the one associated with "oranges". Similarly, if the caption is incomplete (which is often the case, since the caption may not describe every instance in the image), we will ignore those unmatched regions as well. Note that those regions will not be optimized as background, and the loss is only computed with the regions in the matching results.

**Different Strategies for Region-Word Alignment.** Figure 4 shows a comparison to different strategies for the region-word alignment in weakly supervised methods (Zhou et al., 2022; Redmon & Farhadi, 2017; Yao et al., 2021). For fair comparison, we use the region-word alignment loss alone with different strategies for training. "Max-score" (Redmon & Farhadi, 2017; Yao et al., 2021) indicates each word selects the proposal with the largest similarity score, which is widely used in weakly-supervised object detection. However, it tends to assigns each region proposal to a base class due to the full supervision for the bass classes. In contrast, our formulated set-to-set matching aims to minimize the global matching cost to find each region proposal a distinguished corresponding word, where strongly matched base-class regions can push other region proposals to explore the matching with novel classes. "Max-size" (Zhou et al., 2022), assigns all nouns in a caption to the proposal with the max size. The results in Figure 4 show that our method achieves the best performance for the novel classes on both OV-COCO and OV-LVIS.

**Image-Text Alignment.** We conduct ablation experiments to evaluate the effectiveness of different training losses of our method in Table 5. We observe that using the region-word alignment loss alone

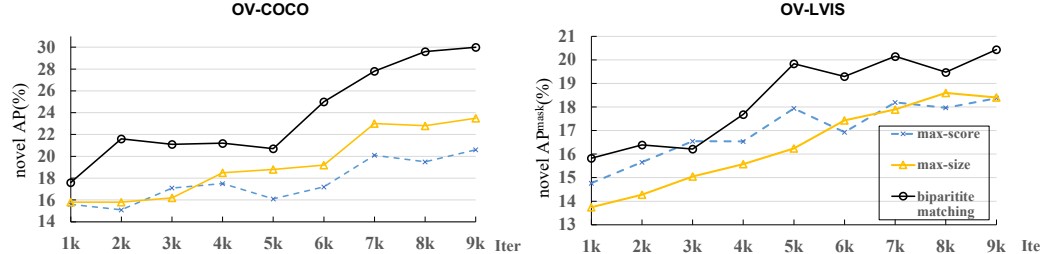

Figure 4: We show a comparison of different alignment strategies in terms of novel-class mAP on both open-vocabulary COCO and LVIS. Different from the bipartite matching that learns the alignment from the global perspective, "max-score" (Redmon & Farhadi, 2017; Yao et al., 2021) does the alignment based on the highest similarity score and "max-size" (Zhou et al., 2022) assigns the max size proposal to the image labels.

Table 5: Ablation study on training losses. It performs best when both the region-word alignment loss and the image-text alignment loss are used, suggesting that they benefit from each other.

| $L_{region-word}$ | $L_{image-text}$ | OV-COCO | | OV-LVIS | |
|:---:|:---:|:---:|:---:|:---:|:---:|
| | | $mAP_{novel}$ | $mAP_{all}$ | $mAP^{mask}_{novel}$ | $mAP^{mask}_{all}$ |
| ✓ | | 30.0 | 44.6 | 20.4 | 30.4 |
| | ✓ | 21.0 | 43.8 | 17.4 | 30.4 |
| ✓ | ✓ | 32.0 | 45.8 | 21.7 | 30.1 |

achieves promising results. It demonstrates the effectiveness of the bipartite matching strategy for learning region-word alignment. Furthermore, we can find that using image-text pair as additional supervision can improve the performance. Intuitively, the image-text pairs data can provide contextual information which complements semantics beyond nouns. A similar conclusion can be found in (Zhou et al., 2022; Zhong et al., 2022).

## 4.6 TRANSFER TO OTHER DATASETS.

To evaluate the generalization ability of our model, we conduct the experiments on transferring COCO-trained model to PASCAL VOC (Everingham et al., 2010) test set and LVIS validation set without re-training. We directly use the model trained with COCO Caption data for OV-COCO and replace the class embeddings of the classifier head. Directly transferring models without using any training images is challenging due to the domain gap. What's more, LVIS includes 1203 object categories which is much larger than COCO label space. As shown in Table 6, our model achieves 2.5 % and 2.0% improvement on VOC test set and LVIS validation set, demonstrating the effectiveness of our method on diverse image domains and language vocabularies. While LVIS includes category names that do not appear in COCO Caption concepts, our model can learn close concepts, which facilitates the transfer to LVIS.

Table 6: Transfer to other datasets. We evaluated COCO-trained model on PASCAL VOC (Everingham et al., 2010) test set and LVIS validation set without re-training. All results are box $AP_{50}$.

| Method | PASCAL VOC | LVIS |
|:---:|:---:|:---:|
| OVRCNN | 52.9 | 5.2 |
| PB-OVD | 59.2 | 8.0 |
| Our | 61.7 | 10.0 |

## 5 CONCLUSIONS AND LIMITATIONS

We have presented VLDet which aims to learn the annotation-free object-language alignment from the image-caption pairs for open-vocabulary object detection. Our key idea is to extract the region-word pairs from a global perspective using bipartite matching by converting images into regions and text into words. We only use a single object detection model supervised by the free image-text pairs without any object-language annotations for novel classes. The extensive experiments show that our detector VLDet achieves a new state-of-the-art performance on both open-vocabulary COCO and LVIS. We hope this work can push the direction of OVOD and inspire more work on large-scale free image-text pair data. The potential limitations of VLDet are: 1) it does not consider the bias of vision-and-language data; 2) we have not investigated even larger data volumes, e.g. Conceptual Captions 12M (Changpinyo et al., 2021). We leave them for future works.

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
