# OpenReview forum: "Learning Object-Language Alignments for Open-Vocabulary Object Detection"
_ICLR.cc/2023/Conference — ICLR 2023 poster_

### Official Review · Reviewer_cyAh · 2022-10-24

**Confidence:** 5
**Correctness:** 3
**Technical Novelty And Significance:** 2
**Empirical Novelty And Significance:** 2
**Recommendation:** 5

**Clarity, Quality, Novelty And Reproducibility:**

**Clarity**

This paper is clear and reads easily. But I still have some questions.
- The last three lines of paragraph "multi-modal object detection" is not accurate.

- How to use image-text caption is unclear. Do you extract the global image features by global pooling? And on which layer? More details are needed.

- I am confused about the experiments of Table 3. If you only use category names, do you still use captions? If so, how? And there are only 6750 nouns for CC-3M dataset? And how to interpret those numbers? More details are required.

- What's the row of using nothing in Table 5?

- Do you generate the pseudo labels online? Do you use a student-teacher framework as in many semi-supervised and weakly-supervised object detection papers?

**Novelty**

Overall, I think the novelty is small, given using weak supervision and bipartite matching are already in the literature.

**Reproducibility**

Although the proposed algorithm is conceptually very simple, I am concerned about if we can reproduce the same improvements, given the detailed ablations are missing. Also the paper doesn't mention the code will be released.

**Strength And Weaknesses:**

**Strength**

+Since the object-language annotation is expensive, it is reasonable to use weak supervision to improve OVOD performances.

+The use of bipartite matching is reasonable and simple.

+The improvements on COCO and LVIS seem nontrivial.


**Weaknesses**

-Using weak supervision to improve OVOD performance is not new, e.g. in RegionCLIP, GLIP, X-DETR, Detic. But those works only show some minor improvements, unlike 5 points gain on LVIS in this paper. I don't see anything fundamentally different, so I am not sure why the gains in this paper are so big.

-Using bipartite matching for WSOD is not new, as seen in [1]. However, this paper doesn't discuss it.

-This paper only shows the results on a base backbone, e.g. ResNet-50. However, many works have already shown high OVOD performances using some large models, e.g. [2]. I am wondering if the gains of this paper can be generalized to large models.

-It seems using weak data can easily get 4-5 points improvement, but I don't see the ablations studies of getting those improvements step-by-step

[1] Omni-DETR: Omni-Supervised Object Detection with Transformers, CVPR 2022

[2] Simple Open-Vocabulary Object Detection with Vision Transformers


**Summary Of The Paper:**

This paper proposes a method to use weak supervision (image-caption pairs) for open-vocabulary object detection (OVOD). This is in fact a problem of weakly-supervised object detection (WSOD). In stead of using the max-score or max-region matching in the literature, this paper proposes to formulate the problem of object-noun matching as a problem of bipartite matching problem, which can be solved by Hungarian algorithm. The experiments on COCO and LVIS have shown that this simple method can have big improvements.

**Summary Of The Review:**

The idea is simple and easy to understand, and it gives some big improvements. But the technical novelty is small, no results comparing with the state-of-the-art, and I am concerned about the reproducibility.

---

> ### Author Response · Authors · 2022-11-18
> **Response to Reviewer cyAh**
>
> We thank the reviewer for the detailed reviews. We provide our responses below
>
> **Q1.The difference with previous work.**
>
> Using image-text pairs to improve object detection in open settings is common. The key is how to use the coarse-grained image-text pairs to learn fine-grained region-word alignments. RegionCLIP and GLIP use pre-trained models to generate pseudo labels for images. Given an image, predicting the pseudo label for each region is independent, which might lose the context information and result in noisy pseudo labels , such as assigning all regions to the same category. Moreover, GLIP uses grounding image-text data which contains detailed bounding box annotations. In contrast, our method is designed for image-text pairs without any bounding box annotations or grounding information and obtains good region-word pairs by finding a bipartite matching with minimal global matching cost. Such global matching helps reduce getting noisy region-word pairs.
>
> **Q2.Using bipartite matching for WSOD is not new, as seen in [1].**
>
> We thank the reviewer for this valuable reference. We add it in our revision (Page 2 Section 2). However, there are a few fundamental differences.
>
> 1. The settings are quite different. First, the weak-supervised setting in [1] is restricted to a closed vocabulary, which is trained and tested in the same categories. Our VLDet follows  the open-vocabulary object detection setting, which aims to improve the unknown class generalization ability. Moreover, [1] trains the object detector using image-level labels which still need manual annotation. Our work uses image-caption datasets which are nearly free and easily cover a large variety of objects.
>
> 2. The methods are quite different. [1] relies on a teacher-student framework and bipartite matching is used for the teacher model which is optimized by the exponential moving average (EMA) from the student. Instead, our VLDet uses a single model and is directly optimized by the bipartite matching results.
>
> 3. The intentions are quite different. [1] considers the problem of how to use unlabeled, fully labeled and weakly labeled annotations in single-modality object detection. However, our work aims to learn the object-word alignments for OVOD from large-scale external image-text pairs.
>
>
> **Q3.The gains of this paper when generalized to large models.**
>
> We provide the results with a larger backbone – Swin-base, to prove the scalability of our proposed approach in page 6 Table 2.
>
> **Q4. Ablation studies for getting improvements step-by-step.**
>
> Our work proposes a simple but effective strategy for learning object-language alignment for OVOD. Actually, we have a comparison study for different alignment strategies in Figure 4, which can be detailized in the table below.
>
> |                    | OV-COCO | OV-LVIS |
> |--------------------|---------|---------|
> | max-score          | 20.6    | 18.4    |
> | max-size           | 23.5    | 18.4    |
> | bipartite matching | 30.0    | 20.4    |
>
> **Q5. The last three lines of the paragraph "multi-modal object detection" is not accurate.**
>
> Thanks for pointing this out. We have revised it.
>
> **Q6. More details on using image-text pairs data.**
>
> Thanks for pointing this out, we add the details in the revision. For the image-text loss, we extract the RoI feature for the image by considering the whole image as a specific region and the entire caption feature from the text encoder as a specific word. In a minibatch, we align each image to its corresponding caption where treating its caption as a positive sample and other captions as negative samples.
>
> **Q7. Details on using category names and nouns in CC3M.**
>
> The comparisons in Table 3 refer to whether we extract category names or nouns from the captions. If we use category names as vocabulary, we do a text matching to see if a category name is in a caption. For example, for a caption of ‘person holds a flag as the shuttle passes by’, only ‘person’ and ‘flag’ will be extracted as words for bipartite matching. If we use all nouns in the caption, ‘person’, ‘flag’ and ‘shuttle’ will be extracted as words.
>
> We follow the RegionCLIP to extract nouns, which parsed the CC3M dataset caption and filtered out the nouns whose frequency is lower than 100.
>
> **Q8. What's the row of using nothing in Table 5?**
>
> In Table 5, we evaluate the effectiveness of different training losses in our method and the results suggest that they benefit from each other. If using nothing, which means not  using our proposed method, then it degenerates to the existing strategies based on max-score or max-size (please refer to Q4).
>
> **Q9. Do you use a student-teacher framework or generate the pseudo labels online ?**
>
>  No, our model neither relies on a student-teacher framework nor generates pseudo labels.
>
> **Q10. I am concerned about the reproducibility.**
>
> We upload our code anonymously with a detailed readme. This code will be released once the paper is accepted.

---

> > ### Comment · Reviewer_cyAh · 2022-12-03
> > **Re: Response to Reviewer cyAh**
> >
> > I would like to thank the authors for clarifying some details of this submission, doing more experiments, updating the paper and sharing the code. However, I am not convinced by the response on the novelty given GLIP, RegionCLIP, Omni-DETR.
> >
> > - I think RegionCLIP uses pre-trained CLIP to generate pseudo labels, but GLIP doesn't use that kind of pre-trained models. GLIP uses mutli-modal attention, so the predictions are not independent. On those image-text data, e.g. CC3M, GLIP doesn't use box-text data. Although it uses some box-caption data from Visual Genome/RefCOCO, etc, this submission also uses box-category data from LVIS/COCO. So I don't think this is a problem.
> > - I am not saying on the paper wise, this submission is similar to Omni-DETR. But this specific idea to use bipartite matching for pseudo labeling is almost the same as this submission. And the problem is in fact a WSOD problem, even though the application is OVOD. So I think the novelty is kind of small.
> > - I am asking is to design some ablations to understand why bipartite matching works so well here, instead of copying some comparison with max-score/max-size from the original submission again.
> > - I would suggest to add the baseline (using nothing) to table 5.
> > - Although some newer numbers are added with swin-base, they still have some gaps from the state-of-the-art.
> > - Even though you use bipartite matching, it is still a pseudo-labeling problem. So the method in fact generates pseudo labels.
> >
> > I think this paper is simple and clean, but the novelty is kind of small. Overall I still think it is marginally below the ICLR threshold.

---

> > > ### Author Response · Authors · 2022-12-07
> > > **Response**
> > >
> > > We thank the reviewer for the detailed reviews and for recognizing our efforts in the revision. For further questions, we give our responses below.
> > >
> > > **1. Compared with GLIP.**
> > >
> > > Although GLIP uses multi-modal attention for vision-language deep fusion instead of CLIP, it still relies on the “max-score” strategy when generating pseudo-labels for image-text data like CC3M. This max-score strategy tends to assign high-confidence region-word pairs which may lead to overfitting to base classes. Besides, GLIP heavily relies on large grounding data. To collect rich semantic grounding data, GLIP uses fully supervised Object365, Flickr30K, VG Caption, and GQA datasets.  These annotated data cover a large vocabulary, which achieves good performance on COCO, while we can see that further using the “max-score” strategy for unlabeled image-text pairs data may lead to a slight drop in performance (see GLIP-T (C) vs. GLIP-T in GLIP Table 3). In contrast, we don’t use grounding data. Our VLDet only uses base categories from COCO or LVIS to initialize the localization capability of the model. Although the detection data is a type of bounding box annotation, it is different from and much small than the grounding data used in GLIP.
> > >
> > > **2. Similar with WSOD work.**
> > >
> > > We agree with the reviewer that both the Omni-DETR and our work use bipartite matching  and share some similarities. Bipartite matching is a well-known technique, but applying it in this particular open-vocabulary object detection (OVOD) context is new. The approach is simple, but coming up with such a simple and effective approach is non-trivial since it requires insights and a lot of trying and testing. We explain and show such a global bipartite matching is particularly effective for the noisy and weak object-language alignment in the open-vocabulary object detection setting, which leads to a large performance gain. None of the existing open-vocabulary object detection methods explore such a simple bipartite matching.  Isn’t a simple and effective solution the best in practice?
> > >
> > > As mentioned in our previous response, Omni-DETR is a WSOD method. We agree that WSOD and open-vocabulary object detection share some similarities, but there are also some fundamental differences. Essentially, WSOD is a closed-set and clean-set problem while open-vocabulary object detection deal with an open-set and noisy-set problem. Just like closed-set and open-set image classification, we can’t say that they are the same problem. WSOD has a strong  assumption that each label has at least one corresponding object in the image while OVOD doesn’t. WSOD requires manually labeled image-level annotations, which OVOD faces with noisy image-text data and an unbounded vocabulary of new categories. The key insight of our work is that we find bipartite matching can learn open-vocabulary object-language alignment from the noisy but nearly free data.
> > >
> > > **3. Lacking ablation study on bipartite matching.**
> > >
> > > Our main contribution is to formulate open-vocabulary object-language alignment as a set matching problem and use bipartite matching to extract the region-word pairs. The ablation study on bipartite matching is compared with other strategies like “max-score” and “max-size”. Note that we have to use one of the strategies for the region-word alignments. It would be helpful if the reviewer could point out which exact ablation study is expected and that we are willing to do it.
> > >
> > > **4. Add using nothing in Table 5.**
> > >
> > > Again there must exist a strategy to learn object-language alignment, for example “max-score” in GLIP, “max-size” in Detic and bipartite matching in our work. Using nothing does not allow the model training.
> > >
> > > **5. Not SOTA result.**
> > >
> > > Although we use CLIP as text encoder like most of the OVOD works, the OWL-ViT [2] from google adopts an end-to-end training recipe with billions of image-text pairs from LiT datastes for image-level training and 1.85 million fully-annotated images from OpenImages V4 and Visual Genome for object-level training, achieving strong open-vocabulary detection. However, it requires large amounts of computing sources and data which are not affordable for us, while we believe we have proven that under the  same backbone (ResNet50 and Swin-B) and training data, our method performs the best.
> > >
> > > **6. The method in fact generates pseudo labels.**
> > >
> > > We agree with the reviewer that the bipartite matching in some sense is similar to generating pseudo labels. However, it is fundamentally different from the pseudo label methods in previous works (e.g. RegionCLIP and GLIP), which need to store the high-confidence pseudo-bounding boxes for image-text pairs for further finetuning, and are clear with a two-step training. In contrast, our VLDet  simply optimizes the bipartite matching results at each iteration and “pseudo labels” generated by bipartite matching are dynamic and different at different iterations.

---

### Official Review · Reviewer_y363 · 2022-10-24

**Confidence:** 5
**Correctness:** 4
**Technical Novelty And Significance:** 3
**Empirical Novelty And Significance:** 4
**Recommendation:** 6

**Clarity, Quality, Novelty And Reproducibility:**

- Overall, I think the paper is of high quality, clearly written, and novel. The method seems easy to reproduce, but the authors seem to release code upon acceptance.
- There are few parts of the text that can be improved, though:
  - The first paragraph on page 5 about the image-text loss is not clear to me. My assumption is that the whole caption is represented by one embedding vector and the whole image as one region vector, which constitutes another (correct) match. However, I think the text in this paragraph could also be misinterpreted as actually having some image-text pairs with ground truth region-word alignment (like in a semi-supervised setting), which I assume is not the case given the whole motivation of the work.
  - Also the paragraph on "Object vocabulary" on page 5 is not clear to me, without having read the corresponding paragraph in the ablation study. It might help to use the notation introduced in Section 3.1.
  - The one-to-one vs. one-to-many experiment needs more clarification. First, the intuition for doing this is missing. I assume it relates to my comment above on how to deal with plural nouns in the text. From the definition of the loss function in Eq. 2, I assume the relation here means one (word) to many (regions). As a suggestion for further investigation, would it help to identify if nouns are plural or singular, and then choose the appropriate constraints (alignment algorithms) for mapping to either one or multiple regions?
  - I think it should be made more clear in the beginning the that the model trains from both detection and image-caption data. Otherwise, the reader may wonder how the model creates sensible regions in the first place.
  - Typo on page 8 last paragraph: "leaning" -> "learning"

**Strength And Weaknesses:**

Strengths:
- The topic is interesting
- The paper is well written, easy to follow, figures a good
- The method is easy and simple, but effective
- The experiments are mostly clear and the method performs well
- Ablation studies are reasonable.

Weaknesses:
- The dataset generalization experiment in Section 4.6 from COCO to Pascal VOC is limited. The 20 Pascal VOC categories are a subset of the 80 COCO categories. The question is how many of the 20 categories are part of the 48 base categories. Still, there will likely only be a few novel categories.
- Did you try a different text embedding architecture that was not jointly trained on images and text? Would this make a difference?
- The related work and the baselines can be improved
  - I think paper [A] (concurrent work) can be discussed in the related work section.
  - The paper [B] is another recent example for OVOD that can be discussed and compared with. It uses pseudo labels and gets strong results on COCO, although in a slightly different setting.
  - The reference "Gu et al." is now an ICLR 2022 paper
- I'm missing a more thorough discussion on the region-word alignment loss
  - It would be good to explain in words how the loss deals with incomplete captions, i.e., when some objects captured in the image are not referred to in the text. Looking at the loss and assuming a perfect matching, it seems such regions would just be ignored by the loss.
  - The nouns in other captions of the same mini-batch (W') may contain the matching (positive) word. How likely does this happen? And why does W' not contain other words of the same caption?
  - How do you deal with plural descriptions in a text, e.g., the caption "Two cats near a dog". Again, I think the loss would just ignore one of the regions that contain a cat (assuming a good matching). Still, I think it would be valuable for the paper to discuss such situations in text.

References:
- [A] Scaling Open-Vocabulary Image Segmentation with Image-Level Labels. Golnaz et al. ECCV 2022
- [B] Exploiting Unlabeled Data with Vision and Language Models for Object Detection. Shao et al. ECCV 2022

**Summary Of The Paper:**

This work proposes a novel method for open-vocabulary object detection (OVOD), where a model is trained from two datasets, one containing bounding box annotations for a set of base categories, and another one containing only free-form text descriptions (captions) of imagesf. While prior work typically uses grounded annotations (words in captions are associated with bounding boxes) or distills information from image-level pre-trained vision and language models, this work proposes a loss function to directly leverage image-text pairs (without grounding supervision). The architecture of the detector is similar to existing ones, with the only exception that text embeddings are predicted instead of a fixed-size probability distribution over a known label space. The model trains from both detection data and image-text pairs. The losses are the same as in standard detectors when training from detection data. For image-text pairs, a similarity is computed between each noun in the caption and each predicted region (region proposals). Bipartite matching is then applied on the similarity matrix to find a matching between regions and nouns, which then serves as ground truth for a standard binary cross-entropy loss. Experiments on the open-vocabulary setting for both COCO and LVIS datasets show improvement over prior baselines, specifically for novel categories.

**Summary Of The Review:**

Overall, I think this a high-quality paper with clear impact to the research community on open-vocabulary object detection.

---

> ### Author Response · Authors · 2022-11-18
> **Response to Reviewer y363**
>
> We thank the reviewer for the detailed reviews. We provide our responses below
>
> **Q1. How many of the 20 categories in VOC are part of the 48 base categories?**
>
> 11 categories in VOC are base classes in COCO. We add this in revision, see Page 9 Section4.6. Noted that the cross-datasets experiment evaluated generalization not only for novel classes but also for different image domains.
>
> **Q2. A different text embedding architecture that was not jointly trained on images and text.**
>
> We follow previous works in open-vocabulary object detection, such as ReginCLIP, Detic, and ViLD, using a fixed CLIP text encoder.  Detic study the effect of different text encoder with FastText [1] which is trained on text only and its result shows that CLIP performs better. For our VLDet with different text encodes, we leave it for future work.
>
> [1] Fasttext. zip: Compressing text classification models. arXiv:1612.03651, 2016
>
> **Q3. Related works [A] and [B].**
>
> Thanks for your suggestion. We have added the discussions in the related work. Please see our revision (Page 3 Section 2).
>
> **Q4. Incomplete captions.**
>
> As the object is not mentioned in the text,  it won’t be matched. Our model is designed to learn region-word alignment. The necessary condition is that the region and word both appear in the image-text pair. Those regions not mentioned in the text will be ignored in matching. Note that it will not be optimized as background, since the loss is only computed with the regions in the matching result. We have added this in the revision. See page 8 Discussion.
>
> **Q5. The possibility of the nouns in W' containing the matching word and the reason W' does not contain other words of the same caption.**
>
> For the first question, if a noun in other captions is the same as the matching word, it will be deleted from the negative word list. However, there is still a small possibility that a word is different from the matching word but with the same meaning. At the current stage, VLDet cannot avoid such case. A possible solution could be: We can calculate the similarity of images or captions when sampling to avoid the same meaning words appearing in the same minibatch. For the second question, to avoid imperfect matching during training, especially at the beginning of training, we didn’t include other words in the same caption as negative words.
>
> **Q6. How to deal with plural descriptions in a text.**
>
> Please refer to our response to Reviewer fcRy Q4. In summary, we select the most confident matching result and ignore other regions to obtain high-quality matching results. We have made it clear in our revision.
>
> **Q7. The text can be improved.**
>
> We really appreciate the reviewer for the constructive comments. We have polished our writing in revision. In summary, we have made the following modifications：
> 1. Provide the  implementation details for the image-text loss.
> 2. Use the notation in ‘Object vocabulary’.
> 3. Clarifythe intuition for one-to-one vs. one-to-many experiments.
> 4. Make it clearer in the beginning that the model is trained from both detection and image-caption data.
> 5. Correct the typo on page 8.

---

> > ### Comment · Reviewer_y363 · 2022-11-25
> > **Re: Response to Reviewer y363**
> >
> > Thank you for the detailed responses. I think most of my questions and comments were answered. I have a few additional comments/questions based on that feedback:
> >
> > - The additional discussion on the one-to-many setting on page 8 is great. One thing that could be added is the aspect of "sample-efficiency" and trade-off between recall and precision, in a sense that the one-to-one mapping is more precise, but has less recall than the one-to-many mapping and hence is less "sample-efficient" (i.e., consider the one-to-many mapping were perfect, then with the same amount of images, you would get more associations to learn from)
> > - The reference to Zhao (Exploiting Unlabeled Data with Vision and Language Models for Object Detection. Shao et al. ECCV 2022) in the related work is wrong - it points to Zhong et al.

---

> > > ### Author Response · Authors · 2022-11-28
> > > **Thank you for your constructive comments**
> > >
> > > Dear Reviewer y363,
> > >
> > > Thank you very much for your constructive comments! We will add "sample-efficiency" in revision and fix the reference. Feel free to discuss more if you have any further questions. We are actively available. Thanks again.

---

### Official Review · Reviewer_W8ar · 2022-10-26

**Confidence:** 5
**Correctness:** 3
**Technical Novelty And Significance:** 3
**Empirical Novelty And Significance:** 3
**Recommendation:** 6

**Clarity, Quality, Novelty And Reproducibility:**

Clarity - the paper is written clearly and easy to follow.
Quality - the quality is good overall with some weaknesses as indicated above.
Originality - novelty is good because the set-matching loss hasn’t been shown effective in the joint training setup to my knowledge.
Reproducibility - the authors promise code release and provide some details of implementation.



**Strength And Weaknesses:**

Strengths:
* The idea of joint training with caption data is promising as captions are noisier and more scalable than image tags
* Set matching is an intuitive approach to handle region-word alignment
* The label space of COCO/LVIS are not used in training phase which follows the open-vocabulary detection settings
* Performance on COCO/LVIS are both strong and the margin over Detic baseline is clear
* Ablation studies are informative.

Weakness:
* Method - what are the constraints for region-text matching? Are all regions assigned to words or are all words assigned to regions? I can imagine some words or regions are noisy and do not have good matches.
* Results - the scalability of the proposed approach needs further study in the axis of data size (e.g. CC12M, larger web-scale image-text data) and model size (larger backbones). It'd be very useful if this approach scales well with data and model size. A simple study is to subsample CC3M and observe the scaling benefits.
* Looks like the proposed approach compromises base categories compared to the Detic baseline. Table 5 image-text loss only has the same overall AP suggests this as well. Is there any explanation for this?
* Comparison with DetPro is unfair because DetPro uses ViLD detector as baseline while this work uses Detic CenterNet as baseline. Detic papers report a ViLD detector baseline which is lower. A more fair comparison would be to report a Mask R-CNN detector (ViLD style) baseline on LVIS.
* SOTA on LVIS claim is not well supported because this study focuses on R50 backbone only.
* [minor] how much overhead does the set-matching add to the training time?


**Summary Of The Paper:**

This paper introduces a set-matching approach for open-vocabulary object detection by jointly training on detection and image caption dataset. The approach achieves good results on the existing COCO and LVIS open-vocabulary detection benchmark.

**Summary Of The Review:**

The paper proposes an interesting approach to tackle open-vocabulary detection by set matching loss on caption data. There are some weaknesses in the paper but I think the merits outweigh the downsides and am leaning positive for the paper.

---

> ### Author Response · Authors · 2022-11-18
> **Response to Reviewer W8ar**
>
> We thank the reviewer for the detailed reviews. We provide our responses below
>
> **Q1.Method - What are the constraints for region-text matching?**
>
> Yes, each noun in the caption is assigned a region.  The number of nouns in a caption is limited, usually less than 10. We provide enough regions for Hungarian matching, which assigns each word a unique region.
>
> We agree that there are noisy words in the caption and noisy regions in the image, such as nouns that appear in the caption but not in the image and objects that appear in the image but not in the caption. The key point here is that the global bipartitie  matching can find each noun a good region matching, which has greatly help the model to learn open-vocabulary region-word alignment, and it does not need to find all the good matchings.
>
> **Q2. Results - The scalability of the proposed approach needs further study in the axis of data size and model size.**
>
> We provide the results with a larger backbone – Swin-base, to prove the scalability of our proposed approach in page 6 Table 2 in the revised version.
>
> For data size scalability, we subsample 1% and 10% of CC3M data and conduct ablation experiments to evaluate the impact of data size on our model. Below are the mask AP results of our VLDet with ResNet50 backbone on OV-LVIS. We can see that more image-text training data help improve the detection performance for both novel and base classes.
>
> | OV-LVIS     | AP_novel | AP_c | AP_f | AP_all |
> | ----------- | -------- | ---- | ---- | ------ |
> | 1% (~30K)   | 16.2     | 28.4 | 34.1 | 28.5   |
> | 10% (~300K) | 18.2     | 29.1 | 34.1 | 29.2   |
> | 100% (~3M)  | 21.7     | 29.8 | 34.3 | 30.1   |
>
> **Q3.The proposed approach compromises base categories compared to the Detic baseline.**
>
> Detic uses class names as object vocabulary which does not strictly follow the open-vocabulary setting. This may avoid some noisy pairs, e.g. when some nouns may not appear in the image, but limit the generalization on novelty classes, especially for large datasets like CC3M, see our Table 3. In our work, we consider a more open setting, where we extract nouns from the caption as object vocabulary.
>
> **Q4.Compared with DetPro and ViLD on a Mask R-CNN detector.**
>
> ViLD distills the knowledge from CLIP visual encoder and trains the network from scratch. It trained about 460 epoches. Detic following their setting takes 4 days on 32 V100 GPUs.  Due to the limited time, it’s unaffoardable for us to implement the ViLD-style detector. However, since our comparison with Detic is fair enough, that Detic performs better than ViLD also shows our effectiveness. Besides, DetPro is not strictly following ViLD which is trained from scratch. Instead, they use a self-supervised pre-trained model SoCo [1] which is designed for object detection to innitialize their model. ViLD and DetPro have fundamental differences with our work, where their performance rely on distilling knowledge from CLIP. Instead, we extend the detector on novel vocabulary with image-text pairs. Nonetheless, we are working on an implementation based on Mask R-CNN and we will update the results as soon as it is available.
> [1] Aligning pretraining for detection via object-level contrastive learning. NeurIPS 2021.
>
> **Q5. How much overhead does the set-matching add to the training time?**
>
> As the Hungarian algorithm is highly efficient, it doesn’t bring heavy computation (see the table below). All experiments are conducted on 8 V100 GPUS.
>
> |         | schedule | Detic | VLDet |
> |---------|----------|-------|-------|
> | OV-LVIS | 4x       | 22h   | 26h   |
> | OV-COCO | 1x       | 16h   | 19h   |

---

> > ### Comment · Reviewer_W8ar · 2022-11-30
> > **Thanks for the feedback.**
> >
> > Thank you for the feedback. I appreciate the clarification on noisy region-text matching and like the additional results on scaling data/model capacity. I think it'd be valuable to compare with ViLD and DetPro on the same architecture (Mask R-CNN) detector if possible. In Q5, at first glance it looks like the overhead is nontrivial about 20% ((19 - 16) / 16). Are the training epoch/iteration and batch size the same here? I assume you have to train more for VLDet since you're using captions. I think it'd be best to report the training step time instead to showcase the efficiency (e.g. 1.0 sec per step). This is relatively minor (no need to do it for the rebuttal) and for your future reference only. Other reviewers have concerns about reproducibility (cyAh) and method clarity (fcRy). In my humble opinion, the method description is clear and the reproducibility looks good/reasonable (the authors promised code release upon acceptance). I'm still leaning towards accepting the paper.

---

> > > ### Author Response · Authors · 2022-11-30
> > > **Thanks for your support!**
> > >
> > > Dear Reviewer W8ar,
> > >
> > > Thank you for your support! We will discuss the training step time for future reference. We are encouraged by the comment that recognized our method description is clear and the reproducibility reasonable. Feel free to discuss more if you have any further questions. We are actively available. Thanks again.

---

### Official Review · Reviewer_fcRy · 2022-10-27

**Confidence:** 5
**Clarity, Quality, Novelty And Reproducibility:** The clarity can be improved, especial…
**Correctness:** 2
**Technical Novelty And Significance:** 2
**Empirical Novelty And Significance:** Not applicable
**Recommendation:** 3

**Strength And Weaknesses:**

Strength
+ Performance is slightly better than previous methods

Weakness
- The paper lacks necessary mathematics proof on solving object language alignments learning. How does bipartite matching is optimized end-to-end during training? Hungarian matching is not differentiable. How does it guarantee convergence on learning?
- It also lacks implementation details. For example, in table 4, how does one-to-one or one-to-all affect the matching? In table 5, how does image-text pair loss implement?
- For Table 1, compared with Base-only, the proposed alignment loss downgrades the performance on supervised base categories. What is the reason?
- For Figure 3, the author avoids example with multiple same instances. What will happen when apply to an image "a basket of oranges"?

**Summary Of The Paper:**

This paper proposes to directly learn from large-scale image-text pair data for open-vocabulary object detection.
It formulates object-language alignment as a set matching problem.
The benefit of such approach is to allowing training on image-text pairs without the help of grounding information.
The author conducts experiments on COCO and LVIS to demonstrate its performance on novel categories.

**Summary Of The Review:**

Overall, I feel solving object language alignment without grounding information is the correct way. However, this paper doesn't prove the effectiveness of using bipartite matching nor well explain the details.

After reading the rebuttal, I still don't get enough details on the correctness of the proposed method. Meanwhile, it also raised me another concern on its ability of handling dense detection. Hence, I would not recommend to accpet this paper.

---

> ### Author Response · Authors · 2022-11-18
> **Response to Reviewer fcRy**
>
> We thank the reviewer for the detailed reviews. We provide our responses below
>
>  **Q1.How does bipartite matching is optimized? Hungarian matching is not differentiable.**
>
>
> We would like to kindly highlight that the matching result by the off-the-shelf Hungarian Algorithm is used as the label for the alignment between image regions and words. It does not participate in the loss backpropagation and does not need to be differentiable. This process can be understood as:
>
> (1) label = Hungarian matching(regions, words).detach()
>
> (2) loss = classfication_loss(label, regions)
>
> (3) loss.backward()
>
> Recall that our motivation is to directly train an object detector from image-text pairs without relying on expensive bounding box annotations. Our key insight is that the global set-to-set matching via bipartite matching help extract reliable paired region-word data, even if they are incomplete. After extracting the paired region-word data, the model optimizes the alignment scores between image regions and  the corresponding words found by the bipartite matching results.
>
>
> **Q2. Lack details. In table 4, how does one-to-one or one-to-all affect the matching? In table 5, how does image-text pair loss implement?**
>
> We formulate extracting region-word pairs from image-text pairs as a set matching problem. Table 4 is an ablation study to analyze the impact of different set matching algorithms. The input of the two algorithms contains both the region set and the word set from the image-text pair. Hungarian algorithm considers one-to-one assignment that matches each word in the caption with only one image region with a global minimum loss. Sinkhorn algorithm considers one-to-many assignment that matches each word with multiple image regions. These two set matching algorithms produce two different region-word matching results, which lead to different optimization targets.
>
> As for the implementation details of the image-text loss in Table 5, we extract the RoI feature for the image by considering the whole image as a region and the caption feature from the text encoder. In a minibatch, we align each image to its corresponding caption where we treat its caption as a positive sample and other captions as negative samples  (see page 4 in our revision).
>
>  **Q3. In Table 1, compared with Base-only, the proposed alignment loss downgrades base class performance.**
>
> The main reason is that the image-text pair might bring noise that interferes the base class detection. For example some nouns in the caption might not match with any region in the image. Such noise may cause the base class performance to drop slightly. Please note that this slight drop happens in many other open-vocabulary works, for example, Detic. We consider it as a trade-off between overfitting to base classes and generalization to novel classes.
>
>  **Q4.What will happen when apply to the image "a basket of oranges".**
>
> In VLDet, we employ the Hungarian algorithm to assign each word a unique region. When facing multiple objects of the same class in an image,  such as “a basket of oranges”, we will align the most confident region-word pair and ignore other “oranges” regions. In other words, our design does not require to match all the “orange” regions. As long as there is a good “orange” region being matched with the word, it is sufficient for the network to learn the region-word alignment to help the open-vocabulary detection task. On the other hand, there is another strategy to deal with this situation, i.e., "one-to-many" in Table 4. In the one-to-many matching strategy, the Sinkhorn algorithm will assign a word to multiple regions, which could match the orange word to multiple orange regions. However, it is also likely to introduce more noise pairs. That’s why the one-to-one matching performs better, which matches each word only with one high-quality image region.

---

> ### Author Response · Authors · 2022-12-01
> **Seek for Active Discussion with Reviewer fcRy**
>
> It is unfortunate to see that  our last detailed response did not resolve your concerns. However, we do  hope the reviewer can clearly specify the details of the concerns and discuss them with us.  We appreciate your time and efforts.
>
> 1. The correctness of the proposed method.
>
> Please give the details of which part of our method is incorrect. The correctness and clarity of our method are well-recognized by the other three reviewers: “ the method description is clear” from Reviewer W8ar, “a high-quality paper with clear impact ” from y363, and “easy to understand” from cyAh.
>
> 2. Ability to handle dense detection.
>
> Again, this is a general comment. It is hard for us to figure out what exactly the problem is. We would like to point out that our method during inference is operating similarly  to Faster R-CNN. It can handle dense detection.

---

### Author Response · Authors · 2022-11-18
**Response to all reviewers**

We are encouraged that the reviewers find our idea is in a correct way for open-vocabulary object detection(Reviewer fcRy) and lean positive from it (Reviewer W8ar) and they view our methodology as reasonable, simple and effective (Reviewer y363, cyAh). We have prepared our detailed individual responses listed below to address each reviewer’s concerns as well as an updated manuscript. Below is a summary of the main changes:

1. We have added the implementation details of image-text pair loss in page 4.

2. We have added the discussion on noisy image-text pairs and incomplete captions in page 8.

3. We have added the experiments for a larger backbone – Swin-base, to prove the scalability of our proposed approach in page 6 Table 2.

4. We have added related works of the mentioned references in page 3.

5. We upload our code for reproducibility in supplementary.

---

### Decision · Program_Chairs · 2023-01-20

**Decision:**

Accept: poster

**Justification For Why Not Higher Score:**

Overall, while I believe this paper should be accepted, I do not believe it reaches the level of a spotlight or oral paper given the discussion.

**Justification For Why Not Lower Score:**

Overall, this paper does have some limitations that could be used to justify rejection. However, I believe that the new perspective it adds to the open-vocabulary problem, compared to prior works, would be of interest to the community and could be further built-upon in future works.

**Metareview: Summary, Strengths And Weaknesses:**

This paper tackles the open-vocabulary object detection problem by leveraging an image-text corpus without any grounding annotation. This differs from prior methods that use some form of grounding annotation or distillation from image-level multi-modal models. Specifically, it proposes a simple method to perform bi-partite matching across a similarity function between nouns in the text and regions in the image, and utilizes that in a standard binary cross-entropy training loss. Also unlike prior methods, this method does not require knowing the label space of the "open" datasets since it mines this from the corpus. Results are shown across COCO and LVIS datasets to show improvements in detection, especially on the novel classes.

The reviewers all appreciated the topic, the writing (for the most part, see caveats), simple but effective method, and improvements. However, there were a number of concerns pointed out including 1) Novelty - Several prior works use weak supervision or even similar ideas of bi-partite matching (unsurprisingly given the generality of the problem) including weakly supervised object detection, 2) Scale with respect to the text corpus size and better architectures (e.g. Swin), 3) Some performance sacrifice on the base classes, 4) Use of different detector baselines (e.g. vs. DetPro), 5) Limitations of the one-to-one mapping, and 6) A number of other clarifying questions. The authors provided extensive rebuttals, including significant new experiments showing scalability across data and model size and explanations on how the work is situated with respect to the prior works.

After the rebuttal, the reviewers had an extensive discussions. Some were not convinced especially by the novelty of the method or strength of the results given the simplicity/empirical nature of the method. Others had concerns about the lack of theoretical or other analysis of the way region-text matching works, although of course most prior methods in this subfield also tend to be empirical in nature. In the end, after having read the paper and considered all of the materials (reviews, rebuttals, and discussions), I recommend accepting this paper. While it is true that the paper has some limitations (most notably some sacrifice on the base classes, making it unclear where it stands in the tradeoff compared to prior methods), it offers a different perspective than prior methods which train on a range of data (including with grounding annotations used either in pre-training or during training) and instead considers image-text pairs in a fine-grained way. The method for doing so is simple, as oftentimes deep learning methods tend to be, but the idea for doing it is compelling given that such data also forms the basis of models such as CLIP. The paper also performs "true" open-vocabulary detection (where the label space is not taken from "open" dataset but instead from the collective nouns in the corpus). All of these perspectives will be of significant interest to the community, and given that the authors plan to release the code it could form the foundation upon which further works can build to address some of the weaknesses.

I do recommend that the authors take the significant discussion to heart and further update the paper, for example running some of the experiments mentioned (e.g. aligning with DetPro) and providing a more extensive exposition on the limitations of the method such as when dealing with potential failure cases of one-to-one mappings.

**Note From Pc:**

if the above contains the word "oral" or "spotlight" please see: "oral" presentation means -> notable-top-5% and "spotlight" means -> notable-top-25%. As stated in our emails, we are disassociating presentation type from AC recommendations